Transcription factor specificity protein (SP) family in renal physiology and diseases

Zhou Wei 1
Fang Jiaxi 2
Jia Qingqing 3
Meng Hanyan 1
Liu Fei 1
http://orcid.org/0000-0002-6076-3806 Mao Jianhua 1 maojh88@zju.edu.cn
1 Department of Nephrology, Children’s Hospital, Zhejiang University School of Medicine, National Clinical Research Center for Child Health , Hangzhou, Zhejiang , China
2 Department of Ultrasound, Taizhou Central Hospital , Taizhou, Zhejiang , China
3 Shanghai Institute of Infectious Disease and Biosecurity, Fudan University , Shanghai , China
Wang Jincheng
Electronic publication date: 2025 Jan 20
Publication date: 2025
Volume: 13
Electronic Location ID: e18820
Received 2024 Apr 10; Accepted 2024 Dec 15
Copyright: © 2025 Zhou et al.
Copyright year: 2025
Copyright holder: Zhou et al.
License: This is an open access article distributed under the terms of the Creative Commons Attribution License, which permits unrestricted use, distribution, reproduction and adaptation in any medium and for any purpose provided that it is properly attributed. For attribution, the original author(s), title, publication source (PeerJ) and either DOI or URL of the article must be cited.
License URL: https://creativecommons.org/licenses/by/4.0/

Keywords: Transcription factor specificity protein, Renal physiology, Diabetic nephropathy, Renal ischemia/reperfusion injury, Renal fibrosis, Renal tumor

Funding: Fundamental Research Funds for the Central Universities 226-2023-00056 National Natural Science Foundation of China 82400854 Zhejiang Xinmiao Talents Program 2023R401208 Scientific Research Fund of Zhejiang University XY2023011 Key Research and Development Plan of Zhejiang Province 2021C03079 This work was supported by grants from the Fundamental Research Funds for the Central Universities (226-2023-00056), the National Natural Science Foundation of China (Grant No. 82400854), the Zhejiang Xinmiao Talents Program (2023R401208), the Scientific Research Fund of Zhejiang University (XY2023011) and Key Research and Development Plan of Zhejiang Province (2021C03079). The funders had no role in study design, data collection and analysis, decision to publish, or preparation of the manuscript.

==============================
Dysregulated specificity proteins (SPs), members of the C2H2 zinc-finger family, are crucial transcription factors (TFs) with implications for renal physiology and diseases. This comprehensive review focuses on the role of SP family members, particularly SP1 and SP3, in renal physiology and pathology. A detailed analysis of their expression and cellular localization in the healthy human kidney is presented, highlighting their involvement in fatty acid metabolism, electrolyte regulation, and the synthesis of important molecules. The review also delves into the diverse roles of SPs in various renal diseases, including renal ischemia/reperfusion injury, diabetic nephropathy, renal interstitial fibrosis, and lupus nephritis, elucidating their molecular mechanisms and potential as therapeutic targets. The review further discusses pharmacological modulation of SPs and its implications for treatment. Our findings provide a comprehensive understanding of SPs in renal health and disease, offering new avenues for targeted therapeutic interventions and precision medicine in nephrology.

Background

The central dogma indicates that transcription is the first step of transmission of genetic information (Crick, 1970). The regulation of transcription factors (TFs) is considered as the upstream control of different biological process (Lambert et al., 2018). TFs exert key effects on regulating different molecular functions and are crucial regulators of embryonic development, cell differentiation, cell reprogramming and tissue growth, which is driven from the interplay between transcription and genome conformation (Goos et al., 2022; Stadhouders, Filion & Graf, 2019). For instance, the Nobel Prize winner Shinya Yamanaka first discovered that four key TFs containing Oct3/4, Sox2, c-MYC and KLF4 had the potential to induce the generation of pluripotent stem cells (iPSCs) from adult or mouse somatic cells (Takahashi et al., 2007; Yamanaka, 2007). Overall, TFs play an important role in the biological process.

Transcription factors play a crucial role in kidney physiology and pathology. They regulate gene expression essential for renal functions such as electrolyte balance, acid-base homeostasis, and blood pressure control (Crislip et al., 2023; Guan et al., 2022; Werth et al., 2017). Dysregulation of these factors is implicated in various renal diseases, including chronic kidney disease, tubular interstitial fibrosis, and renal carcinoma. Understanding the mechanisms by which transcription factors influence renal pathophysiology can provide valuable insights into the development of novel therapeutic strategies for kidney disorders.

Recently, a TF family named specificity proteins (SPs) and Krüppel-like factor proteins (KLFs) received considerable attention in the field of nephrology. The specificity protein (SP) transcription factor gene family sharing homology with KLFs belongs to a C2H2 zinc-finger family (Kaczynski, Cook & Urrutia, 2003; Kim et al., 2017). In the past 40 years, studies have demonstrated that SP/KLFs not only regulate physiological processes including proliferation, differentiation, development, etc., but also pathogenic states, including tumorous and inflammatory disorders (Kim et al., 2017). In human, nine SPs and 17 KLFs have been identified, and more details about SP/KLF family proteins can be achieved from the UniProt website (https://www.uniprot.org/). The differences and homolog of amino acid sequences among SP/KLF family members were shown in Figs. 1A–1C, revealing that the SPs may have more complex roles and functions in diseases as they have longer amino acid sequences than KLFs on average.

Figure 1 The comparison between SP family proteins and KLF family proteins.

(A) The overview of all KLF/SP family members and their corresponding amino acid amounts. (B) The guide tree of KLF/SP family members representing amino acid sequence homology among different proteins. (C) The difference of amino acid amounts between the SP family and KLF family. *** means P < 0.001.

The former researchers have reviewed the role of the KLF family in renal physiology and disease (Mallipattu, Estrada & He, 2017; Rane, Zhao & Cai, 2019). However, none has summarized the function of the SP family in the kidney, and the protein sequences of the SP family is quite different from the KLF family based on the above analysis of the phylogenetic tree (Fig. 1B). Additionally, the length of SPs is overall longer than KLFs, especially SP1-4, indicating that the structure and function of the SP family is more complex than the KLF family (Fig. 1C). Interestingly, a recent study has revealed that SP1-4 are also recognized as a TF superfamily with overlapping DNA binding that facilitate chromatin accessibility (Zhao et al., 2022). Therefore, this review firstly focuses on the global understanding of the SP family functioning in renal physiology and diseases. Our review on the intricacies of SPs’ molecular mechanisms in various renal pathologies provides a strong foundation for future research and potential therapeutic interventions.

Survey Methodology

We conducted a literature search on PubMed using the keywords “specificity protein,” “transcription factor,” “kidney disease,” “nephrology,” and “renal disease.” However, articles that were not original research articles or reviews were excluded from our analysis.

The sp family members in normal kidney

In the context of renal physiology, SPs have been shown to regulate the transcription of various genes, including those involved in fatty acid-CoA biosynthesis and Na+-H+ exchange (Muthusamy et al., 2018; Sheng et al., 2005; Wu et al., 2024). SPs exert their regulatory effects by binding to GC/GT boxes in gene regulatory regions and through various post-translational modifications such as phosphorylation, acetylation, glycosylation, and methylation (Sun et al., 2024). These modifications enable SPs to enhance or inhibit gene transcription, modulate protein activity, and regulate the expression of non-coding genes (Li et al., 2023a).

The expression and cellular location of SPs in human healthy kidney

The cluster of nine SP proteins is believed to originate from a common ancestor in both chordates and the broader Eumetazoa (Dailey, Kozmikova & Somorjai, 2017). Concluded from four transcriptome datasets in the PubMed database, there is almost no gene expression of SP7-9 in healthy kidneys; moreover, the mRNA, circular mRNA and protein expression of SP1-4 is significantly higher than SP5-6 in all datasets mentioned, among which the results of SP family gene in the Human Protein Atlas (HPA, http://www.proteinatlas.org/) were shown in Figs. 2A–2E (Duff et al., 2015; Fagerberg et al., 2014; Szabo et al., 2015). Interestingly, SP1, SP3 and SP4 have similar structure features, while SP2 is different from them that it has much higher affinity to GT box than GC box while other SPs have more access to GC box (Hagen et al., 1992; Kingsley & Winoto, 1992). Further, based on the single-cell sequencing results in HPA and Kidney Interactive Transcriptomics (KIT, http://humphreyslab.com/SingleCell/), SP1-3 exist in all kinds of renal cell types, SP4 mainly occurs in proximal tubular cells and immune cells including T, B cell and macrophages, while relatively few SP5 distributes in distal tubular and collecting cells, and SP6 distributes in few proximal cells and collecting cells (Figs. 2F–2K).

Figure 2 The mRNA, protein expression and distribution of SP family members in kidney from human protein atlas (HPA) database.

(A) The mRNA level of SP1-9 from transcriptome data. (B) The protein expression and location of SP1 in human normal kidney tissue. (C) The protein expression and location of SP2 in human normal kidney tissue. (D) The protein expression and location of SP3 in human normal kidney tissue. (E) The protein expression and location of SP4 in human normal kidney tissue. (F) The mRNA expression and distribution of SP1 in different kinds of renal cell types. (G) The mRNA expression and distribution of SP2 in different kinds of renal cell types. (H) The mRNA expression and distribution of SP3 in different kinds of renal cell types. (I) The mRNA expression and distribution of SP4 in different kinds of renal cell types. (J) The mRNA expression and distribution of SP5 in different kinds of renal cell types. (K) The mRNA expression and distribution of SP6 in different kinds of renal cell types. ** means P < 0.01, **** means P < 0.0001.

The function of SPs in renal physiology

In physiological conditions, members of the SP family are involved in the regulation of fatty acid metabolism, water and electrolyte balance, renin generation, and the maintenance of internal environment homeostasis. Fatty Acid Elongase 7 (ELOVL7) is an enzyme catalyzing the rate-limiting step towards the synthesis of very long-chain fatty acids and exhibiting the highest activity toward C18: 3 (n-3) acyl-CoAs and C18: 3 (n-6) acyl-CoAs in humans, which is highly expressed in bovine kidney (Naganuma et al., 2011). Attentionally, the transcription activity of ELOVL7 in bovine mammary epithelial cells is regulated by SP1, and the interaction between SP1 and the ELOVL7 promoter is strengthened by the exogenous α-linolenic acid (ALA, 18: 3n-3) (Chen et al., 2018a). Regarding electrolyte regulation, SP1, SP3 and SP4 can recognize the Na(+)/H(+) exchanger isoform-2 (NHE-2) gene full of GC bases that is rich in mouse inner medullary collecting duct (mIMCD-3) cells. However, only SP1 can activate NHE-2 in mIMCD-3 cells, which can be repressed by SP3 or SP4 (Bai et al., 2001). In addition, SP1 and SP3 are responsible for activating the transcription of alpha-epithelial sodium channel 2 (α-ENaC2), which accounts for a significant portion of α-ENaC transcripts in human lung and kidney tissue (Chu, Cockrell & Ferro, 2003). This activation occurs through a GC-rich element in the promoter region as well. Hepatocyte growth factor (HGF) receptor (c-met) is widely expressed in mouse kidney whose expression is positively relative to the expression of SP1 and SP3 (Zhang et al., 2003). Multiple studies have supported that SP1 and SP3 synergistically bind to the GC boxes in the promoter region of the c-met gene (Papineni et al., 2009; Seol, Chen & Zarnegar, 2000; Zhang et al., 2003). As4.1 cells are a clonal cell line derived from the kidney neoplasm of a transgenic mouse, which express high levels of renin encoded by Ren-1c or Ren2 (Abel & Gross, 1988, 1990), and mutation of SP1/SP3 site reduce Ren-1(c) expression (Pan et al., 2003). Human hyaluronan synthase 2 gene (HAS2) synthesize the linear glycosaminoglycan hyaluronan at the plasma membrane. In human renal proximal tubular epithelial cell line, HAS2 is identified by SP1 and SP3, mediating the constitutive transcription augments (Monslow et al., 2006).

The role of sps in different renal diseases

SP family has been proved to regulate multiple target genes, which can be shown in the TRRUST database that is a manually curated database containing both human and mouse transcriptional regulatory networks (Han et al., 2018). From TRRUST, SP1 has 472 reported target genes and SP3 has 113 target genes in human. Attentionally, SP1 shares 98 known targets with SP3. SP2 and SP4 have nine and 11 targets separately, which are less than SP1 or SP3 (Details can be seen from Table S1). In addition, SP7 has only one known target gene from TRRUST, while SP5, SP6, SP8 and SP9 are not recorded in TRRUST.

The role of SP1 in different renal diseases

As the first member to be discovered among the SP family, SP1 has been studied much more than other members (Dynan & Tjian, 1983a, 1983b), meanwhile, it has more regulatory target genes than other SP family members. It is worth noting that SP1 exhibits an increased glomerular and proximal tubular expression in all forms of human glomerulonephritis (GN), including minimal change glomerular disease, membranous GN, focal segmental glomerulosclerosis, membranoproliferative GN, IgA nephropathy, acute diffuse proliferative GN, pauci-immune GN and so on (Kassimatis et al., 2010).

The role of SP1 in renal ischemia/reperfusion injury

Renal ischemia/reperfusion (I/R) injury often occurs on patients suffering from the perioperative period, resulting in rapid kidney dysfunction and a high rate of mortality (Barrera-Chimal et al., 2015; Granata et al., 2022; Wu, Siow & Karmin, 2010). It was reported that significant reductions appeared in both SP1 and its downstream regulatory Klotho levels in renal tubular epithelial cells (RTECs) subjected to I/R injury (Hu et al., 2010; Huang et al., 2021; Li et al., 2020). Furthermore, exogenously administered SP1 or Klotho demonstrated individual protective roles in the context of H/R injury (Qian et al., 2018; Yuan et al., 2017). Although the total protein of SP1 in I/R kidney is similar to that in the sham-control kidney, serine-phosphorylation of SP1 was remarkably elevated in the kidney subjected to I/R, which is activated via the phosphorylated ERK and microRNA-204 (Chen et al., 2017; Wu, Siow & Karmin, 2010; Yang et al., 2021a). SP1 also binds to and activates the promotor region of sphingosine kinase (SK) 1 directly in HK-2 cells, subsequently increasing sphinganine-1-phosphate (S1P) formation, which alleviates renal I/R injury (Yuan et al., 2017). Besides, the mRNA and protein expression of folate transporters including folate receptor 1 (FOLR1) and folate carrier (RFC) are decreased under I/R stimulation, which can be downregulated by SP1 in tubular cells (Yang et al., 2021a). Notably, folate is an essential micronutrient for human body. Altogether, the above evidence mentioned suggests that SP1 is tightly associated with I/R injury, indicating that SP1 activation may be beneficial for I/R-associated acute kidney injury.

The role of SP1 in diabetic nephropathy

Diabetic nephropathy (DN) is a prominent cause of end-stage renal disease (ESRD), characterized by interstitial fibrosis and glomerular sclerosis (Flyvbjerg, 2017; Webster et al., 2017). Recent research has extensively examined the role of SP1 in the pathogenesis and progression of DN. It has been discerned that its primary influence on DN development predominantly stems from its capacity to modulate inflammation, oxidative stress, and pro-fibrogenesis. SP1 has also been implicated in regulating high glucose (HG)-induced gene expression in the aortic endothelial cells (Du et al., 2000). Besides, bioinformatics analysis reveals that SP1 may be related to diabetic nephropathy and WNT/β-catenin pathway (Hu et al., 2021; McKay et al., 2016). The protein expression of SP1 is significantly up-regulated in the renal cortical tissue of diabetic db/db mice, and further elevated with the aging of db/db mice (Chen et al., 2014; Li et al., 2023b). In both in vivo and in vitro settings, Zhang et al. (2017) have found that SP1 expression significantly increases in podocytes under diabetic conditions. However, another study shows that mRNA and protein expression of SP1 are both decreased in MPC5 cells under the HG treatment (Zhang et al., 2021). Additionally, advanced glycation end products-modified bovine serum albumin (AGE-BSA) have been found to inhibit Neuropilin-1 (NRP1) promoter transcriptional activity in podocytes by reducing the binding capacity of the SP1, affecting the development of DN (Bondeva & Wolf, 2009; Bondeva & Wolf, 2015). Furthermore, Jeong has introduced a novel molecular strategy using ring SP1 decoy ODNs delivered via HVJ-liposome gene-delivery technique, showing potential in preventing and treating diabetic nephropathy (Kang et al., 2008). Consequently, targeting SP1 may emerge as a viable therapeutic strategy for managing diabetic nephropathy. In diabetic nephropathy, persistent hyperglycemia activates NF-κB, which assumes a central role in inflammation. Chen et al. (2014) observed an interaction between elevated SP1 and NFκB-p65 within the podocyte nucleus upon exposure to high glucose. Furthermore, they identify the direct binding between SP1 and the DNMT1 promoter region, indicating the SP1/NFκB p65-Dnmt1 pathway may be a therapeutic target for protecting against podocyte injury in DN (Zhang et al., 2017).

Abundant evidence suggests that oxidative stress is heightened and mitochondrial dysfunction is exacerbated in diabetic nephropathy, ultimately resulting in renal injury (Forbes & Thorburn, 2018; Hallan & Sharma, 2016). Zhang et al. (2021) have proposed that SP1 can bind to the three SP1 response elements within the peroxiredoxin 6 (Prdx6) promoter, thereby directly influencing the transcriptional activation of Prdx6 in podocytes. The SP1-mediated upregulation of Prdx6 expression serves as a protective mechanism against podocyte injury in diabetic nephropathy by mitigating oxidative stress and inhibiting ferroptosis.

SP1 plays a crucial role in orchestrating key downstream processes associated with extracellular matrix (ECM) accumulation in high-glucose conditions or hyperglycemia. In such environments, there is an upregulation of the tubular-specific enzyme myo-inositol oxygenase (MIOX) (Sharma et al., 2017). Electrophoretic mobility shift assays have demonstrated a robust binding of the SP1 to the MIOX promoter, particularly with the unmethylated SP1 oligo, under high-glucose conditions. Treatment with SP1 siRNA leads to a reduction in mRNA and protein levels of both SP1 and MIOX, as well as the generation of reactive oxygen species originating from NADPH oxidase (NOX)-4 and mitochondrial sources. This intervention also leads to a decrease in hypoxia-inducible factor-1α (HIF-1α) and fibronectin expression, an extracellular matrix protein known to be elevated in diabetic nephropathy and tubulopathy. Additionally, Gao et al. (2016) have proposed that c-Jun and SP1 are pivotal upstream regulators of TGFβ1 activation during the fibrotic progression of DN. It has been reported that the renoprotective factor nuclear factor erythroid 2-related factor 2 (NRF2) can reverse TGFβ1-mediated glomerular fibrosis, but only in the early stages of DN (Jiang et al., 2010). Nrf2 delays the progression of DN by repressing TGFβ1 in a c-Jun and SP1-dependent way (Gao et al., 2014; Gao et al., 2016).

SP1 is also implicated in the dysregulation of multiple RNAs in diabetic nephropathy. For example, Chen et al. (2018b) noted increased expression of long non-coding RNA ZFAS1, which was activated by SP1. Yang et al. (2021b) showed that SP1 overexpression in diabetic nephropathy increases mitochondrial RNAase P (Rmrp) levels, which sponges miR-1a-3p to promote JunD expression. Furthermore, bioinformatics assays, luciferase reporter assays, and chromatin immunoprecipitation (ChIP) assays confirmed that SP1 could bind to the promoter region of Rmrp (Yang et al., 2021b). Piyush also found a significant reduction in the expression of miR29b in a model of diabetic nephropathy in renal proximal tubular epithelial cells (RTECs) exposed to high glucose. At the same time, there was an observed upregulation in the expression of DNA methyltransferases (DNMTs), specifically DNMT1, DNMT3A, DNMT3B and SP1. Further investigation revealed that miR29b targets DNMT1 by regulating its transcription factor, SP1 (Gondaliya et al., 2018).

Thus, modulating expression of SP1 may serve as a therapeutic option to treat diabetic nephropathy.

The role of SP1 in renal interstitial fibrosis

SP1, a widely distributed transcription factor, plays a pivotal role in various pathophysiological contexts, encompassing cell cycle regulation, apoptosis, angiogenesis and embryogenesis (Chae et al., 2006; Zhang et al., 2003). The protein level of SP1 is markedly increased in the unilateral ureteral obstruction (UUO) kidney (Kim et al., 2013). The occurrence and progress of renal fibrosis usually attribute to three or more cell types, including the injury of renal tubular epithelial cells, activation of myofibroblasts, proliferation of mesangial cells, and infiltration of immune cells (Humphreys, 2018; Tang, Nikolic-Paterson & Lan, 2019; Zhao, 2019). Anomalies in the function of RTECs constitute a central factor contributing to aberrant tubular reabsorption and renal fibrosis (Liu et al., 2019; Yao et al., 2017). Notably, SP1 has been previously implicated in regulating CD2AP promoter activity and expression in RTECs, underscoring its functional role in this cell type (Lu et al., 2008). Electrophoretic mobility shift assays (EMSA) conducted with nuclear extracts from HK-2 cells have identified SP1 and multiple SP3 isoforms, whose co-mutation suppresses the gene transcription of macrophage, which can mediate the progression of renal fibrosis (Hughes et al., 2002; Li, Fu & Liu, 2022). Furthermore, the transition of quiescent glomerular mesangial cells (MCs) into a highly proliferative phenotype with myofibroblast-like characteristics is a common process in inflammatory renal glomerular diseases, ultimately leading to glomerulosclerosis (Liu et al., 2023). This transition is mediated by the membrane-associated enzyme membrane type 1 (MT1)-MMP (Turck et al., 1996). Alfonso-Jaume, Mahimkar & Lovett (2004) reported that MT1-MMP transcription in glomerular mesangial cells is cooperatively regulated by nuclear factor of activated T cells (NFAT) c1, along with the zinc finger transcription factors SP1, SP3, and Egr-1.

As is known to all, TGFβ1/Smad signaling is a classic pathway closely associated with renal fibrosis. It is noteworthy that specific binding to TGFβ1 promoter region is identified by SP1 (Geiser et al., 1993). Synthetic double-stranded oligodeoxynucleotides containing consensus sequences that bind to specific transcription factors can block mRNA transcription at the DNA level. Sung et al. (2013) found that SP1 decoy ODNs reduced SP1-DNA and Smad-DNA complexes in a murine model of UUO-induced renal fibrosis, decreasing TGF-β1, IL-1β, p-Smad, and type I collagen expression. Consistent with this, a new chimeric (Chi) ODN method also support that simultaneous inhibition of the transcription factors NF-κB and SP1 attenuates tubulointerstitial fibrosis in a UUO model (Kim et al., 2013). Caveolin-1 (cav-1), an integral membrane protein, plays a crucial role in the synthesis of matrix proteins by glomerular mesangial cells (MC) (Fujita et al., 2004). Follistatin, a widely expressed and secreted glycoprotein, counteracts the profibrotic and proinflammatory actions of several TGFβ-1 isoforms, particularly activins (de Kretser et al., 2012; Hedger & de Kretser, 2013). The absence of cav-1 results in increased nuclear levels of SP1, which regulates the activation of the follistatin promoter in cav-1 knockout MCs. This occurs through enhanced phosphoinositide 3-kinase (PI3K) activity and downstream protein kinase C (PKC) zeta-mediated phosphorylation (Mehta et al., 2019). In addition, many biological processes mediated by TGFβ-1 rely on SP1 in cooperation with Smad proteins, such as Smad3 and Smad4 (Datta, Blake & Moses, 2000; Wu et al., 2007). For instance, SP1 collaborates with Smad proteins to mediate TGFβ-1-induced fibrogenesis and is essential for the production of type I collagen induced by miR-29c downregulation in NRK-52E cells (Jiang et al., 2013). Recent research also has shown that p-Smad2/3 interacts with SP1 and p300 to transmit TGFβ-1 signals in human glomerulonephritis (Kassimatis et al., 2010). Interestingly, the deletion of Smad7 promotes Ang II-induced upregulation of SP1 in hypertensive kidneys. The enhanced SP1-TGFβ1/Smad3-NF-κB signaling pathway may be the primary mechanistic route by which the deletion of Smad7 exacerbates Ang II-mediated renal fibrosis and inflammation (Liu et al., 2013). Furthermore, Li et al. (2020) reported that SP1 overexpression mitigated TGFβ1-induced fibrosis in HK-2 cells by inducing Klotho expression. SP1 was found to directly regulate Klotho expression in kidney cells by binding to a specific CG-rich site within the Klotho promoter region. This finding provides valuable insights into the transcriptional regulation of Klotho in renal disease models.

Overall, SP1 is an important pro-fibrotic target for renal fibrosis, and many treating strategies based on SP1 has been developed until now.

The role of SP1 in lupus nephritis

Lupus nephritis (LN) is a prominent manifestation of systemic lupus erythematosus (SLE), which has an impact on morbidity and mortality (Mackay et al., 2006; Shen et al., 2024). Despite advancements in management, as many as 20% of patients ultimately progress to end-stage renal disease (ESRD) (Almaani, Meara & Rovin, 2017). In the early stages of renal fibrogenesis, evidence from exosomal urinary miRNA and cellular pellets derived from patients with biopsy-proven SLE indicates that miR-29c interacts with SP1 to regulate collagen production in tubular epithelial cells (Solé et al., 2019). Conversely, miR-150 and miR-21 expression contribute to the perpetuation and amplification of the fibrotic process leading to ESRD. This occurs through a Smad3/TGFβ pathway, independent of SP1, during later stages of LN progression. Axl, a member of the TAM (Tyro-3, Axl, and Mer) family of receptor tyrosine kinases (RTKs), exerts influence over various disease states, including lupus-like autoimmune disease (Cohen et al., 2002; Lu & Lemke, 2001), which are strongly upregulated in both mouse models of nephritis and streptozotocin-induced diabetic nephropathy (Yanagita et al., 2002; Zhen, Priest & Shao, 2016). In the context of mesangial Axl expression regulation, Zhen et al. (2018) identified the transcription factor SP1 as a crucial regulator.

The role of SP1 in renal cell carcinoma

Growing evidence indicates that specificity protein transcription factors, including SP1, SP3, and SP4, are overexpressed in tumors and play a crucial role in regulating genes essential for cancer cell death and survival (Chadalapaka et al., 2010, 2008). SP1 is highly expressed in most tumors and has a malignant phenotype, whose abnormal activation can be involved in the pathogenesis of many tumors (Black, Black & Azizkhan-Clifford, 2001; Chang & Hung, 2012). In renal cell carcinoma (RCC), Zhu et al. (2020) found that the knockout of the SP1 gene significantly inhibits cell proliferation and induces cell cycle arrest at the G1 phase. Liu et al. (2020a) identified that the VHL/HIF-2α/SMYD3 signaling cascade mediates EGFR hyperactivity in RCC, and SP1 works with SMYD3 to promote EGFR expression and amplify its downstream signaling activity. Disrupting SP1 accumulation at the EGFR locus can downregulate SMYD3, thereby inhibiting tumor growth. Clear cell renal cell carcinoma (ccRCC) is the most common subtype of renal cell carcinoma (RCC), accounting for 70–80% of all RCC cases (Zhang et al., 2024). Wang et al. (2024) indicated that ubiquitin B regulated the expression of VEGFA in a SP1-dependent manner, thereby modulating the angiogenic capability of RCC cells. ChIP-qPCR assay confirmed that SP1 binds to the promoter region of FBXL6, and FBXL6 depletion restrains ccRCC progression. So, targeting the regulation of SP may provide new insights for exploring targeted therapies for ccRCC based on F-box proteins and substrates (Yu et al., 2022). In another report, it was also demonstrated that SP1 induces upregulation of lncRNA SNHG14 as a competitive RNA, thereby promoting the migration and invasion of ccRCC by modulating N-WASP (Liu et al., 2017).

The role of SP2 in renal diseases

Little has been reported about SP2 in the kidney. SP2, a member of the SP family, is widely expressed but has limited DNA binding or transcriptional activity in human and mouse cell lines, and remains relatively poorly characterized (Xie et al., 2010). A bioinformatics analysis reveals that vascular endothelial growth factor A (VEGFA) and its regulatory TF SP2 exert important effects on the development of clear cell sarcoma of the kidney (Wang et al., 2016). However, some roles of SP2 have been described in other diseases. As the same as SP1, SP2 also participates in TGFβ-1/Smad signaling in the osteogenic differentiation of valvular interstitial cells (Zheng et al., 2022). Additionally, SP2 is an inherited factor required for embryonic development in zebrafish lines (Xie et al., 2010). Furthermore, SP2 activates Ski-mediated astrocyte proliferation under the stimulation by LPS (Da et al., 2021). Therefore, SP2 may play a role in the growth and differentiation of kidney.

The role of SP3 in renal diseases

SP3 is firstly reported as a SP1 inhibitor in SL2 cells (Hagen et al., 1994). SP3 plays multiple pivotal roles in cell function and homeostasis (Suske, Bruford & Philipsen, 2005), whose function is usually reported together with other members of SP family. For example, combined overexpression of v-Myc and SP1/SP3 leads to oncogenic transformation of fibroblasts (McCormick & Maher, 2011), while SP1, SP3, and SP4 have all been reported to bind to vascular endothelial growth factor receptors (VEGFRs) promoters and increase vascular endothelial growth factor (VEGF) receptor expression in pancreatic cancer cells (Abdelrahim et al., 2007). Bioinformatics analysis on several online databases confirms that SHMT1 and SHMT2 may be potential therapeutic and prognostic biomarkers in patients with RCC and SP3 may be important a target for RCC treatment (Situ et al., 2023).

Moreover, SP3 exhibits a close connection with renal fibrosis. Deng, Xu & Huang (2022) have reported a co-regulatory network involving Pax2, Pax5, SP1, SP2, SP3, SP4 along with Bckdha-dependent miR-125a-3p and VEGFa-dependent miR-199a-5p, which plays a pivotal role in tightly controlling Bckdha/VEGFa in the pathogenesis of renal fibrosis across both mice and humans. The calcium-sensing receptor (CASR) is related to multiple kidney diseases, such as calcium kidney stones, chronic kidney disease and heavy metals-induced nephrotoxicity, whose activity is stimulated almostly 2–3-fold induced by IL-6 in proximal tubule HKC cells (Canaff, Zhou & Hendy, 2008; Kosiba et al., 2020; Torres & De Broe, 2012; Vezzoli et al., 2011). Oligonucleotide precipitation assay verifies that the promoter region is regulated by SP1/3 and Stat1/3 together (Canaff, Zhou & Hendy, 2008). The ubiquitous mammalian extracellular matrix glycosaminoglycan, hyaluronan (HA), plays a crucial role in regulating cell phenotype in fibrosis and scarring. HA is synthesized at the cell membrane by hyaluronan synthase (HAS) enzymes, encoded by the HAS1–3 multigene family in humans (Abdelrahim et al., 2007; Monslow et al., 2003). Prior in vitro studies have suggested that both the transcriptional induction of HAS2 and subsequent HAS2-driven HA synthesis may contribute to kidney fibrosis through phenotypic modulation of RTECs (Lewis et al., 2008; Rouschop et al., 2004). Recent work by Daryn on HAS2 mRNA sequences has shown that the widely expressed zinc finger transcription factors SP1 and SP3 bind to multiple recognition sites upstream of the transcription start site, mediating constitutive HAS2 transcription (Monslow et al., 2006). In addition, TGF-β1 and IL-1β have been found to upregulate HAS1 in dermal fibroblasts and renal proximal tubular epithelial cells. Using siRNAs to knockdown transcription factor mRNAs, Chen et al. (2012) demonstrated that TGF-β1-induced upregulation of HAS1 transcription is mediated via Smad3, whereas HAS1 induction by IL-1β relies on SP3 dependently. Thus, SP3 is also a pro-fibrotic gene, which usually cooperates with SP1.

The role of SP4 in renal diseases

Apart from some bioinformatics research, little is known about SP4 in kidney. Most research involving SP4 concentrate on the central nervous system (CNS), as it is highly expressed in embryos in the developing CNS (Ramos et al., 2009; Saia et al., 2014; Supp et al., 1996). SP4 enhances peritoneal angiogenesis regulated by Interleukin-6/soluble IL-6 receptor by increasing vascular endothelial growth factor production in peritoneal dialysis (Catar et al., 2017; Zhu et al., 2023). From Figs. 2A, 2E and 2I, the expression and distribution of SP4 is less than SP1-3. Hence, we speculate that the function of SP4 in kidney is weak.

The role of SP5 and SP6 in kidney

Little is known about SP5 or SP6 in kidney. Their expression is very low in human kidney under normal physiological conditions (Figs. 2A, 2J and 2K). However, some literature shows that both SP5 and SP6 participate in some important biological signaling, which is also closely related to renal physiology and disease.

SP5 is different from any other member of the SP1 family with little homology, and it is dynamically expressed in the early stage of embryo development (Harrison et al., 2000). The expression of SP5 is elevated in several human digestive system tumors, which is also a direct target by WNT in human pluripotent stem cells (Chen et al., 2006; Huggins et al., 2017; Katoh, 2018; Takahashi et al., 2005). In the embryonic mouse telencephalon, activation of Wnt/β-catenin signaling leads to the up-regulation of SP5 (Fujimura et al., 2007; Treichel, Becker & Gruss, 2001). During the mouse central nervous system development, SP5 competes with SP1 to directly bind to P21, mediating the negative feedback responses to Wnt/β-catenin signaling (Fujimura et al., 2007; Hoverter et al., 2012; Treichel, Becker & Gruss, 2001). Additionally, SP5 and SP8 are gene-specific transcriptional coactivators in the Wnt/β-catenin pathway in differentiating embryonic stem cells and mouse embryos (Kennedy et al., 2016).

SP6, which is also called KLF14 or epiprofin, was firstly identified in 2000 (Scohy et al., 2000). SP6 has been proved to a master regulator of tissue development, including the apical ectodermal ridge of limb buds, posterior neuropore, teeth and hair follicles of embryos (Haro et al., 2014; Nakamura et al., 2004; Rhodes et al., 2021; Smith et al., 2020). SP6 regulates multiple extracellular matrix genes in the developing tooth, including COL1A2, COL11A2, HALPN1 (Rhodes et al., 2021). SP6 and SP8 synergistically form and maintain the apical ectodermal ridge, functioning as indispensable mediators of Wnt/β-catenin and Bmp signaling in the limb ectoderm (Haro et al., 2014). Recently, SP6 has been proved to determine both proliferation of the inner enamel epithelium and its differentiation into ameloblasts (Smith et al., 2020).

All the above findings illustrate that SP5 or SP6 functions as a regulator of cell growth, indicating either SP5 or SP6 may serve as an important role in the early development of kidney.

Pharmacological modulation of sps/therapeutic applications of sps

The participation of SP/KLF proteins in modulating signaling pathways during processes such as inflammation, oxidative stress, abnormal cell proliferation, and fibrosis has been subject to in-depth investigation and extensive review across molecular and cellular levels. Rane, Zhao & Cai (2019) have summarized the therapeutic applications of KLFs in renal diseases. The various SPs discussed in this review could serve as therapeutic drug targets in treatment of specific kidney diseases.

Research has shown that SP1 regulates the levels of leptin during the maintenance of internal stability, and this regulation is triggered by the metabolic process of glucose spurred by insulin and fatty acid metabolism (Moreno-Aliaga et al., 2007). For example, SP1 increases the activity of genes associated with metabolism, such as ATPase copper transporting alpha (ATP7A) and apolipoprotein A-1 (APOAI) (Georgopoulos et al., 2000; Xie & Collins, 2013). SP1 is also reported to regulate expression of the sodium-glucose co-transporter (SGLT) in the rabbit intestinal epithelium (Kekuda, Saha & Sundaram, 2008). Empagliflozin, an inhibitor of the sodium-glucose co-transporter 2 (SGLT2) utilized for managing type 2 diabetes mellitus, effectively reduces blood glucose levels and mitigates inflammation by hindering renal glucose reabsorption. Liu et al. (2020b) reported that empagliflozin improves diabetic renal tubular injury by mitigating mitochondrial fission through the AMPK/SP1/PGAM5 pathway. Luciferase reporter assays and ChIP confirmed SP1 binding to the PGAM5 promoter at −539 nt, increasing PGAM5 expression and worsening diabetic renal tubular injury (Liu et al., 2020b). Thus, empagliflozin may alleviate DN by inhibiting SP1 indirectly. Melatonin, a pineal hormone derived from serotonin, has an increasing body of evidence supporting its therapeutic role in cancer prevention, treatment, and delaying the onset of various cancer subtypes (Mehrzadi et al., 2020; Stepniak et al., 2022). Woo et al. (2022) found that ovarian tumor domain-containing protein 1 (OTUD1) knockdown inhibited melatonin-induced Bim deubiquitination at the lysine 3 residue and apoptosis in cancer cells. In addition, melatonin-induced activation of SP1 was found to be involved in OTUD1 upregulation at the transcriptional level, and pharmacological inhibition and genetic ablation of SP1 (siRNA) interrupted melatonin-induced OTUD1-mediated Bim upregulation (Woo et al., 2022). In addition, Licochalcone A was also discovered to promote autophagy and LC3B expression, induce cell cycle arrest, and inhibit the migration and invasion of RCC cells by modulating FAK/Src signaling pathway-mediated SP1 expression. The effects of LC3B on the metastatic phenotype of ACHN cells was enhanced with the overexpression of SP1 (Tseng et al., 2023).

Liver fibrosis and kidney fibrosis share common or similar pathological features and mechanisms, which manifest as abnormal accumulation and excessive deposition of ECM. The pathogenic process involves the activation or transdifferentiation of ECM-producing cells, such as myofibroblasts, in response to stimuli like lipid peroxidation products, cytokines, and an abnormal extracellular matrix environment during pathological injury. Clinically, it has been confirmed that non-alcoholic fatty liver disease (NAFLD) is an independent risk factor for the rapid development of chronic kidney disease (CKD) (Roderburg et al., 2023). Within the realm of liver disorders, SP1 and SP3 have a role in promoting liver fibrosis produced by leptin by increasing the production of collagen type I, alpha 1 chain (COL1A1) (García-Ruiz et al., 2012). It has also been reported that SP1 can interact with Smad3 to enhance TGF-β-induced fibrotic responses and SP1 is necessary for Ang II–induced fibrotic and inflammatory response (Datta, Blake & Moses, 2000; Liu et al., 2013). A selective SP1 inhibitor named plicamycin has been developed (Choi et al., 2014). Plicamycin specially targeting SP1 protein in the M2 macrophage can significantly inhibit acupuncture-induced fibrosis of ligamentum flavum in rats via reducing cross-talk between fibroblasts and M2 macrophages (Feng et al., 2023). Thus, plicamycin may also be an effective drug for treating renal fibrosis. Current research suggests that the excessive accumulation of ECM in the glomeruli is a significant pathological alteration in FSGS (Barisoni, Schnaper & Kopp, 2007). Matrix metalloproteinases (MMPs) are crucial enzymes in regulating the degradation of ECM, participating in the remodeling of ECM and stabilizing tissue development and the internal environment. Among them, MMP-9 primarily degrades type IV collagen, and tissue inhibitor of metalloproteinase-1 (TIMP-1) serves as a specific inhibitor for MMP-9. The dynamic balance between MMP-9 and TIMP-1 plays a key role in the physiological equilibrium of ECM (Catania, Chen & Parrish, 2007). Moreover, MMP9 is transcriptionally activated by SP1 (Wang et al., 2018). Therefore, targeting SP1 may reduce the occurrence and development of FCGS.

Conclusion and challenges

In the current review article, we mainly highlight the role of SP1 and SP3 in kidney diseases as the other members of SP family protein is less expressed in kidney tissue (Figs. 2A–2K). Recent studies on various signaling proteins (SPs) have significantly expanded our understanding of their diverse biological and physiological functions, as well as their involvement in numerous organs, tissues, and cellular processes. While these proteins typically serve important biological functions, aberrant expression or dysfunction of SPs has been implicated in various disease processes. Consequently, despite earlier research emphasizing the roles of SPs in development and proliferation, it is now evident that these proteins play a crucial role in the progression of several diseases, including kidney diseases. Targeting SP1 or SP3 may exert positive effects on renal diseases. We also conclude the function of other SP family members in Table 1.

Table 1 The summary of SP family members in renal physiology and diseases.

Renal physiology	Renal diseases	
SP1 regulates fatty acid metabolism, water-electrolyte balance, renin expression, and hyaluronic acid synthesis.	SP1 alleviates renal ischemia-reperfusion injury and promotes renal fibrosis, diabetic nephropathy, and lupus nephritis. Furthermore, it also facilitates the pathological processes of tumors.	
SP3 regulates water-electrolyte balance and renin expression.	SP3 cooperates with SP1 to enhance renal fibrosis.	
The role of SP4 in the kidney may be limited.	
SP4 regulates water-electrolyte balance.	SP2, SP5 and SP6 may regulate cell proliferation and renal development.	

While a role for SPs in kidney diseases has been identified, there is a need to understand the mechanisms that control SPs expression and function in renal physiology and disease. SP1, SP2, and SP3 are ubiquitously expressed in various renal cell types, making it challenging to target SPs for therapeutic applications in specific tissues. Additionally, due to the redundancy of SP family members and their common transcriptional targets, it is critical to identify specific SP protein binding partners, accessory proteins, and downstream gene targets in specific cell types and diseases. Intervention at the binding interface between transcription factors and specific DNA promoter sequences is expected to become a promising direction for the development of drugs targeting the SP family. These studies will provide mechanistic insights into specific SP functions in different tissues, leading to the identification of novel therapeutic targets and paving the way for precision medicine approaches.

Supplemental Information

Supplemental Information 1 Downstream Regulatory Gene Information of the SP Family Recorded in the TRRUST Database.

Additional Information and Declarations

Competing Interests

Author Contributions

Data Availability

The authors declare that they have no competing interests.

Wei Zhou conceived and designed the experiments, analyzed the data, authored or reviewed drafts of the article, and approved the final draft.

Jiaxi Fang performed the experiments, prepared figures and/or tables, and approved the final draft.

Qingqing Jia conceived and designed the experiments, analyzed the data, authored or reviewed drafts of the article, and approved the final draft.

Hanyan Meng performed the experiments, prepared figures and/or tables, authored or reviewed drafts of the article, and approved the final draft.

Fei Liu performed the experiments, authored or reviewed drafts of the article, and approved the final draft.

Jianhua Mao conceived and designed the experiments, analyzed the data, prepared figures and/or tables, authored or reviewed drafts of the article, and approved the final draft.

The following information was supplied regarding data availability:

This is a literature review.

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
