# Peer review of "Transcription factor specificity protein (SP) family in renal physiology and diseases"

_PeerJ, doi:10.7717/peerj.18820_

## Round 0.1 · original submission · Major Revisions

1. The title could be more concise and focused on the main content of the review, such as "Transcription factor Specificity protein (SP) family in renal physiology and diseases".
2. The abstract should provide a more refined and focused summary of the article's core content and highlights. The concluding statement about innovative therapeutic strategies is too general.
3. The introduction's background on the SP transcription factor family is slightly lengthy and could be more focused on kidney-related content. The lead-in to each subsection is insufficient.
4. The discussion of the role of SP transcription factors in renal physiology mainly focuses on SP1 and SP3, with limited coverage of other members like SP2 and SP4, lacking comprehensiveness.
5. In the renal disease section, the level of detail in describing the roles of SP transcription factors varies across different diseases. For example, the discussion of SP1 in diabetic nephropathy is very detailed, while it is relatively brief in lupus nephritis and other conditions.
6. There is a lack of systematic summary and in-depth discussion of the mechanisms of action of SP transcription factors in the kidney, failing to well reveal the inherent patterns and differences in their regulation.
7. In the conclusion and outlook section, the discussion on the pharmacological modulation of SP transcription factors and their potential as therapeutic targets could be more thorough.
8. Some sentences in the text need to be more concise, as they are a bit lengthy. A few words are misspelled and need correction, such as "Krupsilonppel-like".

Reviewer 1 ·

Basic reporting

The manuscript titled "The Role of SP Family Transcription Factors in Renal Diseases" effectively discusses the involvement of Sp1, Sp2, Sp3, Sp4, Sp5, and Sp6 transcription factors in renal diseases, particularly in renal fibrosis and lupus nephritis, shedding light on their potential as therapeutic targets. However, I would like to suggest some revisions to further enhance the clarity and impact of this work:
1. While the paper effectively highlights the roles of SP family transcription factors in kidney diseases, providing more mechanistic insights into how these factors regulate gene expression and interact with specific signaling pathways would greatly strengthen the manuscript.
2. The comparison between SP family proteins and KLF family proteins is intriguing. Expanding on this comparison and discussing the unique roles or interactions of SP family members in renal diseases compared to other transcription factor families would be beneficial.

Experimental design

no comment

Validity of the findings

1. Emphasizing the clinical relevance of targeting SP family transcription factors in the management of renal diseases would make the paper more impactful. Discussing ongoing clinical trials or potential therapeutic strategies targeting these factors could be valuable.
2. Enhancing the discussion section to provide a more comprehensive overview of the current understanding of SP family transcription factors in renal physiology and pathology would be beneficial for readers seeking a deeper insight into this topic.

Additional comments

1. Including a graph illustrating the pathways or mechanisms through which SP family transcription factors contribute to renal diseases could aid in visualizing complex concepts for the readers.

Reviewer 2 ·

Basic reporting

no comment

Experimental design

no comment

Validity of the findings

no comment

Additional comments

the authors summarize the current knowledge of SPs in renal physiology and diseases, paving the way for innovative therapeutic strategies and precision medicine approaches. This review impresses me with its clear and well-organized structure, coupled with a robust logical flow. The meticulous attention to detail contributes to the overall strength of the review, making it both insightful and informative. However, several small concerns may be addressed in the revision before publication.

1. This review delves the diverse roles of SPs in various renal diseases, including their participation in renal ischemia/reperfusion injury, diabetic nephropathy, renal interstitial fibrosis, and lupus nephritis. However,The topic of this review is transcription factor speciûcity protein (SP) family in renal physiology and pathology. Since kidney diseases include primary nephropathy, secondary nephropathy, urinary tract infections, acute kidney injury, kidney stones, and carcinoma of kidney. There might be limited literature on kidney stones, but carcinoma of kidney represents a significant area for discussion. My suggestion is for the author to include a discussion on carcinoma of kidney.

2.
It appears that the format of some references is incorrect, such as Reference 17. Please revise it using the correct format. Be careful.

I commend the authors for their comprehensive review, which exhibits a commendable level of organization and clarity. Overall, the authors have made a valuable contribution to the field, and with some refinements, this review could serve as an even more impactful resource.

---

## Round 0.2 · Minor Revisions

Many thanks for authors' revision. This mauscript has been improved.

Please address reviewer 1's comments.

For Figure 1, authors can describe more in figure legends instead of labeling.

Reviewer 1 ·

Basic reporting

1. The manuscript is written in clear and professional English, with terminology appropriate for an expert audience in nephrology and molecular biology.
2. The manuscript provides a thorough overview of the role of Specificity protein (SP) transcription factors, specifically SP1 and SP3, in renal physiology and diseases. The references are recent and relevant, supporting the discussion effectively. However, the authors could enhance the introduction by clarifying clearer why a specific focus on the SP family, rather than other transcription factor families, offers unique insights into renal pathophysiology.
3. Figures effectively illustrate the SP family members' roles and their molecular pathways, enhancing comprehension. However, a few figures (e.g., depicting gene expression variations in different renal cells) could benefit from higher resolution or clearer labeling for improved accessibility.

Experimental design

NA

Validity of the findings

NA

Additional comments

1. The introduction briefly covers transcription factors' general role in renal physiology, but could expand on why the SP family is particularly relevant for kidney diseases. A short section contrasting SPs with other major TF families involved in renal pathology (e.g., NF-κB, AP-1) could contextualize the significance of this review.
2. The detailed account of SP1, SP3, and their regulatory roles in fatty acid metabolism, electrolyte balance, and gene expression in kidney cells is well done. However, including a comparison table or figure summarizing the functional differences between SP1, SP3, and other SPs in the kidney could help readers assimilate this information more quickly.
3. The conclusion rightly emphasizes the critical role of SP1, SP2 and SP3 in kidney disease. However, it would benefit from a summary of major unresolved questions, such as the redundancy within the SP family, and potential biomarkers that could predict responsiveness to SP-targeting therapies.

Reviewer 2 ·

Basic reporting

no comment

Experimental design

no comment

Validity of the findings

no comment

Additional comments

Thanks to the author for the revision, I think it meets the criteria for publication

---

## Round 0.3 · accepted · Accept

Authors have addressed all the reviewers’ concerns. The review firstly summarizes the important role of transcription factor SP family in renal physiology and diseases. Different from common reviews, the review collects the gene information from the public databases, and compare the literature summary with these bioinformatics data. Overall, the paper provides a new style for writing a review and highlights the importance of specific SP TFs in kidney. I think this paper can be accepted for publication.